# Aircraft-based observation of volatile organic compounds (VOCs) over the North China Plain

Yibo Huangfu[1,#], Ziyang Liu[1,#], Bin Yuan[1,*], Sihang Wang[1], Xianjun He[1], Wei Zhou[2], Fei Wang[2], Ping Tian[2], Wei Xiao[2], Yuanmou Du[2], Jiujiang Sheng[2,*], Min Shao[1]

[1] College of Environment and Climate, Institute for Environmental and Climate Research, Guangdong-Hongkong-Macau Joint Laboratory of Collaborative Innovation for Environmental Quality, Jinan University, Guangzhou 511443, China

[2] Beijing Weather Modification Center, Beijing 100089, China

[#] These authors contributed equally to this work

[*] Corresponding author: Bin Yuan (byuan@jnu.edu.cn), Jiujiang Sheng (jiujiangsheng@163.com)

**Abstracts.**

The vertical distribution of reactive trace gases can greatly help understand the complex atmospheric evolution under the joint impacts of surface emission, chemical removal, and regional transport. Focusing on the core area of the North China Plain, aircraft-based observations were conducted in September 2017 and July 2019 to reveal the vertical distributions of volatile organic compounds (VOCs) measured by high-time resolution mass spectrometry. Generally decreasing trends of VOC concentrations with altitudes were captured, indicating strong surface source emissions and chemical removal within the planetary boundary layer (PBL). Ethanol exhibited the highest concentration within the PBL with an average of 46.7 ppbv and the largest ratio (16.5) between the average below and above the PBL heights. The vertical-averaged VOCs above Baoding were greater than those in Beijing by factors ranging from 1.2 to 3.5, suggesting richer precursors for secondary pollutant formation in Baoding. Increases of several VOC species, including styrene and acetonitrile, at high altitudes (>2500 m) were captured in Beijing. Correlation analysis further revealed the significant influences of industrial and biomass burning emissions. Our results highlight the critical role of both local emissions and regional transport in shaping the VOC vertical distributions, which may affect atmospheric organic chemistry across various atmospheric layers in the region.

**Keywords**: Vertical profiles; aircraft-based observation; volatile organic compounds

## 1 Introduction

The North China Plain (NCP) stands as one of the most developed city clusters in China, yet it is also a region of concern that suffers from severe air pollution (Zhao et al., 2021;Yao et al., 2022b;Le et al., 2020;Wang et al., 2023). Following the implementation of the Action Plan on the Prevention and Control of Air Pollution, the particulate matter concentrations in NCP have significantly declined primarily due to a sharp reduction in anthropogenic emissions. In contrast, ozone levels have not shown a similar downward trend (Chen et al., 2020;Li et al., 2019a;Lu et al., 2019;Lu et al., 2020). From 2013 to 2019, the annual maximum daily average of 8-h ozone (MDA8O$_3$) concentrations increased significantly with a rate of 2.3 ppbv yr$^{-1}$ in Beijing (Chen et al., 2020), while during warm seasons (Apr. – Sep.), the average increasing rate reached 3.3 ppbv yr$^{-1}$ for NCP (Lu et al., 2020). Under the background of coordinated efforts to reduce pollution and carbon emissions, the continuous improvement of ambient air quality in the NCP region is under great pressure.

Volatile organic compounds (VOCs), as the key precursors of ozone and secondary organic aerosols due to photochemical degradation, have been listed as one of the key indicators for atmospheric environmental quality during the "14[th] Five-Year Plan" period (Li et al., 2022b;Mao et al., 2021). The temporal variation, emission characteristics, and environmental effect of VOCs have been extensively studied in the NCP region through ground-based measurements, greatly enhancing the understanding of the role in the formation of photochemical pollution (Yuan et al., 2012;Wang et al., 2014;Li et al., 2019b;Zhao et al., 2021;Sun et al., 2018;Forster et al., 2023). However, recent studies have also revealed the complex evolution of VOCs in the atmosphere under the impact of meteorological conditions (Zhang et al., 2018;Shi et al., 2018) and inter-city and regional transport (Liu et al., 2019;Zhao et al., 2021;Chang et al., 2019) that cannot be thoroughly investigated based on ground measurements.

In the past few years, vertical observations have been carried out utilizing observation towers (Zhang et al., 2020;Li et al., 2022a;Yang et al., 2024a;Li et al., 2024), tethered balloons (Zhang et al., 2019;Sangiorgi et al., 2011;Zhang et al., 2018), unmanned aerial vehicles (UAVs) (Peng et al., 2015;Han et al., 2016;Vo et al., 2018), and aircraft (Benish et al., 2020;Forster et al., 2023;Zhao et al., 2021;Wilde et al., 2021), bringing the opportunity to study VOCs from a three-dimensional perspective. Compared to other vertical observation techniques, aircraft platforms offer distinct

advantages, including the capability of covering larger spatial areas, carrying heavy instrument payloads, and performing both vertical and horizontal observations at higher altitudes (Kwak et al., 2020;Zhao et al., 2021;Dieu Hien et al., 2019). The advantages of aircraft measurement make it an irreplaceable role in scientific research and practical applications. While aircraft-based research has been conducted intensively in the United States and Europe with high temporal resolution instruments (Fisher et al., 2016;Karion et al., 2015;Ren et al., 2018), only a handful of similar aircraft flights have been conducted equipped with online instruments that measure inorganic pollutants and aerosols in China (Zhao et al., 2021;Liu et al., 2018). Limited aircraft-based VOC measurements in China typically applied the collection of canister samples followed by detection by gas chromatography techniques in the laboratory (Xue et al., 2011;Benish et al., 2020;Liu et al., 2013). A typical decrease of non-methane hydrocarbons with increasing height was reported over Northeast China (Xue et al., 2011), while for the NCP region, aircraft-based measurements have been conducted but only reported the vertical distributions of BTEX species (benzene, toluene, ethylbenzene, xylenes), showing a similar negative trend with height (Liu et al., 2013). Nevertheless, the vertical distribution of VOCs in the NCP region is still unclear due to the scarcity of offline samples, hindering the ability to accurately assess the impacts of local emissions and air mass transport on VOC levels.

This study analyses aircraft-based observation results over the Beijing and Baoding area, the two core cities of the NCP region. A proton transfer reaction time-of-flight mass spectrometer (PTR-ToF-MS) was equipped on the platform to monitor VOC concentrations with high time resolution across altitudes. Combined with correlation analysis of VOC pairs, possible sources and origins of VOCs within the planetary boundary layer (PBL) and the atmosphere above it were discussed, pointing out the important roles of local urban emissions and regional transport.

## 2 Methods

### 2.1 Aircraft flight routes

A King Air 350ER aircraft (Hawker Beechcraft) was deployed and five flights were conducted between Sep. 9[th] and 15[th] in 2017, and an additional flight on July 14[th], 2019. The flight trajectories are summarized in **Table 1** and illustrated in **Figure 1**. The aircraft was based at Shahe Airport (40°8'24" N, 116°19'48" E) in Changping District, Beijing, surrounded by villages, factories, and educational institutions. Four of the

flights in 2017 (Sep. 9$^{th}$, 12$^{th}$, 13$^{th}$, and 15$^{th}$) focused on the Beijing area, while the
remaining two flights (Sep. 14$^{th}$, 2017, and Jul. 14$^{th}$, 2019) followed southwest
trajectories over the forest area before turning southeast to the Baoding area and
returning to Shahe Airport. During each flight, the aircraft maintained level flight at a
given altitude before ascending or descending to the next predetermined flight level.
The step intervals between altitudes were consistently maintained within approximately
150 to 200 meters.

## 2.2 VOC sample and meteorological data collection

107       A PTR-ToF-MS (PTR-ToF-MS-8000, Ionicon Analytik GmbH, Austria) was

applied onboard to measure VOC concentrations during each flight. The time-of-flight
mass spectrometry has been proven to have the capability of measuring VOCs at high
time resolution over broad mass spectra (Wu et al., 2020a;Yuan et al., 2017). Ambient
air was drawn through a 1.5-m-long PTFE tube at a flow rate of 15 L/min using a pump.
A sub-stream of this air was then subsampled by the PTR-ToF-MS at a flow rate of 100
mL/min through a PTFE membrane particle filter. With a voyage speed of ~250 km/h
of the aircraft, VOC concentrations were measured every 5 s, resulting in a spatial
resolution of ~70 m. The PTR-ToF-MS was operated with a drift pressure of 2.2 mbar,
E/N 135 Td, and reactor temperature of 60 ℃. The raw spectral data of PTR-ToF-MS
were processed using the Ionicon Data Analyzer (IDA, V2.0.1.0), including mass
calibration, peak detection, peak fitting, and etc. The sensitivities of PTR-ToF-MS for
various VOC species were calibrated with commercial standard gas (Apel-Riemer,
Environmental Inc., USA) before each field campaign. The sensitivity of monoterpenes
was calibrated based on the a-pinene in the standard gas. Both methyl vinyl ketone
(MVK) and methacrolein (MACR) were included in the standard gas, so the sensitivity
of MVK&MACR was calculated based on their summed concentrations. A total of 15
VOC species are reported in this study and listed in **Table S1** as well as the limits of
detection (LODs) and propagated uncertainties. Based on the multi-level tests in the
laboratory, the impacts of humidity on VOC sensitivities were evaluated to be less than
10% for the reported VOC species, so no correction was conducted, and the induced
uncertainty was propagated to the overall uncertainties. The interferences of
fragmentations, such as the impact of higher-carbon aldehydes and cycloalkanes on
isoprene signal (m/z 69, $C_5H_8H^+$) and the impact of ethylbenzene on benzene signal
(m/z 79, $C_6H_6H^+$), were not corrected, so the concentrations of isoprene and benzene
might be overestimated due to these interferences.
In addition, the aircraft platform also carried instruments that recorded the
meteorological parameters (AIMMS-20, Aventech Research Inc.), including
temperature, relative humidity, and pressure. The vertical profiles of meteorological
factors are shown in **Figure S1**. A global positioning system (GPS) was equipped to
record the aircraft's position. All instruments were thoroughly inspected and air-tightly
checked before each flight to ensure the quality and reliability of data. To minimize the
impact of aircraft exhaust emissions on VOC measurement, the data measured during
the first 2 minutes after the engine started were excluded from the profile data.
**2.3 Height of planetary boundary layer**
In this study, the height of planetary boundary layer (HPBL) was determined by
the air parcel method (Zhao et al., 2020;Zhao et al., 2019), which was considered to be
more accurate and less deviated by human subjective judgment compared with the
visual observation method and data simulation method (Zhang et al., 2019). The air
parcel method determines the HPBL based on the vertical profile change of the potential
temperature ($T_\theta$) during the adiabatic process (Zhang et al., 2018). The vertical profiles
of potential temperature for each flight can be found in **Figure S1**. $T_\theta$ first decreases
with height to a minimum and then increases. The HPBL is determined as the height
where $T_\theta$ returns to its surface value. The HPBL during each ascending and descending
stage is listed in **Table S2**. A 10% uncertainty is determined based on previous studies
(Zhao et al., 2019;Zhang et al., 2014;Vogelezang and Holtslag, 1996).
**3 Results and discussion**
**3.1 The characteristics of VOC vertical distributions in Beijing**
The ambient air quality data of criteria pollutants (ozone, $NO_2$, $SO_2$, CO, $PM_{2.5}$,
and $PM_{10}$) of Changping Town station, the nearest national air quality monitoring
station to the Shahe Airport (11 km away), were collected from the China National
Environmental Monitoring Center (CNEMC) and presented in **Figure 2**. According to
the Level-II thresholds in China's current National Ambient Air Quality Standards
(NAAQS), pollution events in Beijing with $PM_{2.5}$ exceedances on Sep. 9[th], 10[th], and
14[th] and ozone exceedances on Sep. 13[th] were noticed. The vertical profiles of relative
humidity (RH), temperature (T), and $T_\theta$ in **Figure S1** provide meteorological
information for each flight. Monotonic trends for T and $T_\theta$ were recorded. T decreased
with altitude, while $T_\theta$ had opposite trends. In contrast, RH showed a more complex,
non-monotonic trend with altitude, with great variations found across different aerial
surveys, reflecting the complexity of the boundary layer structure.

Five flights were conducted over Beijing in Sep. 2017. The averaged

concentrations of all 15 VOC species ranged from $25.9 \pm 13.4$ ppb measured on Sep.12[th],
2017 to $52.1 \pm 57.7$ ppb measured on Sep.14[th], 2017 (**Table S3**). The VOC
concentrations measured within and above the PBL are listed in **Tables S4 and S5**. As
illustrated in **Figure 3**, the VOC concentrations within the PBL (**Table S4**) were
compared with previous measurements in urban Beijing (Squires et al., 2020;Yuan et
al., 2012). Yuan et al. (2012) conducted VOC measurements on the top of a six-story
building on Peking University campus using the PTR-MS technique in the summer of
2010. The concentrations of aromatic species below the PBL in this study were
comparable to those measured in 2010 (**Figure 3b**), with data points clustering along
the 1:1 line. For the OVOC species, the differences remained within or close to twofold
variations. The two datasets in **Figure 3b** were measured seven years apart, during
which VOC emissions in urban Beijing declined significantly(Wang et al., 2015;Yao et
al., 2022a). While the VOC levels at the campus site in 2010 should have been higher
due to greater VOC emissions, this effect was likely compensated for by the extra
industrial emissions in suburban areas, which might be the reason for the comparable
results observed.

VOC measurements by PTR-ToF-MS in the summer of 2017 were conducted at

the 102 m platform of the Institute of Atmospheric Physics (IAP) meteorology tower
and represented the VOC concentrations driven by traffic-related emissions in the
center of urban Beijing (Squires et al., 2020). Compared with the 2017 IAP
measurement, the aromatic species measured in this study were higher by factors
ranging from 3.2 to 6.2 (**Figure 3c**). The enhancements of C8 and C9 aromatics are
much greater than those of benzene and toluene. Given that industrial emissions contain
higher proportions of C8 and C9 aromatics (Wang et al., 2024;Jiang et al., 2023), these
results suggest that the VOC measured in aerial surveys in this study might be under
the impacts of industrial emissions from the suburban region, especially at lower
altitudes. Most OVOC species measured across both campaigns showed good
agreement within twofold variability, except methyl ethyl ketone (MEK), which
exhibited a factor of 5.7 higher in this study. As one of the common ingredients in
industrial solvents, MEK can be emitted through multiple industrial processes (Wu et

al., 2020b;Wang et al., 2024). Similarly, tetrahydrofuran (THF), an isomer of MEK and a significant industrial pollutant itself (Hu et al., 2018), may also contribute to the measured MEK signals. Hence, the higher concentrations of both compounds in this study are likely attributable to the nearby industrial emissions. Regarding the biogenic species, isoprene, MVK&MACR, and monoterpenes measured in this study were consistently higher than those reported in the other two campaigns, probably due to the enhanced biogenic emissions from the suburban region surrounded by mountain vegetation (**Figure 1**).

The characteristics of VOC vertical distributions were investigated through ten profiles obtained across the five flights over Beijing in 2017. Composite profiles of individual VOC species, as shown in **Figure 4**, revealed fundamental vertical distribution patterns, with detailed profiles for each flight provided in **Figures S2-S6**. Generally, decreasing trends with altitudes were observed for most VOC species, and the largest concentration variations were noticed towards the surface due to the dynamic change of surface emissions. These general decreasing trends of pollutants have been reported by previous studies (Benish et al., 2020;Liu et al., 2013;Xue et al., 2011). However, due to a lack of high-resolution VOC measurements, concentration variations with height and anomalous enhancement have not been documented.

All aromatic hydrocarbons exhibited characteristic negative vertical gradients with maximum concentrations at ground level (0.3-4.5 ppbv by average) and progressive decreases through the PBL, except the anomalous profile of styrene (**Figure 4**). Aromatic hydrocarbons primarily originate from vehicular and industrial emissions. and their lifetime within the PBL can span several days. The dominance of surface emissions explains their significantly higher concentrations near the ground than those measured above the the PBL. As turbulent mixing transports air upward, these compounds get oxidized with OH radicals, diminishing to near-zero levels at higher altitudes with minimal variation. The ground levels of C8 aromatics were the highest, reaching 4.5 ppbv on average. In contrast to other aromatic hydrocarbons, styrene displayed a distinct increase with altitude above the PBL, peaking notably at ~3500 m. Similar concentration enhancements at the same altitude were also observed for methanol, acetonitrile, and acetaldehyde. By checking the profiles of each flight (**Figures S2-S6**), such enhancements in the composite profiles were mainly driven by the measurement on Sep. 9[th], 2017. This anomalous profile, potentially associated with long-range transported industrial emissions, will be further explored in **Section 3.3**.

Similar VOC concentration enhancements were also found on the flight for Jul. 14th,
2019, which will be analyzed as well in **Section 3.3**.

OVOCs originate from both biogenic and anthropogenic sources and can also be
formed as secondary products during the oxidation of non-methane hydrocarbons,
complicating the interpretation of their vertical distributions (Yuan et al., 2012). Near
the surface, OVOC concentrations were substantially higher than those of aromatic
hydrocarbons. Across all flights, the vertical profiles of both alcohols and MEK
followed trends similar to aromatic hydrocarbons. Their vertical variations demonstrate
the impact of PBL dynamics, as evidenced by their pronounced vertical gradients
(**Figure 4**). During the flight on Sep. 12th, the HPBL over Beijing exhibited the largest
disparity between ascending and descending stages, with a difference of 700 m (**Table
S2**). As illustrated in **Figure S3**, when the HPBL increased from ~900 m to ~1600 m,
turbulent mixing transported ground methanol and ethanol upward, leading to a
significant increase in alcohol concentrations above 1000 m. In contrast to other
OVOCs, acetone and acetaldehyde displayed markedly greater variability above the
PBL on Sep. 12th. Both compounds are known to form secondarily through atmospheric
oxidation (Holzinger et al., 2005;de Gouw et al., 2004;Wu et al., 2020a), which likely
contributes to their less uniform vertical distribution and larger variability at higher
altitudes.

For biogenic VOCs, isoprene and monoterpenes are both primarily emitted from
biogenic sources, while MVK and MACR are secondary oxidation products of isoprene
(Canaval et al., 2020;Cappellin et al., 2019). Across all flight measurements, the vertical
distribution patterns of MVK and MACR were quite similar to those observed for
alcohols and most aromatic hydrocarbons. In contrast, the vertical profiles of isoprene
and monoterpenes varied significantly between flights. Decreasing trends with altitude
were found on Sep. 9th (**Figure S2**), Sep. 13th (**Figure S4**), and Sep. 14th (**Figure S5**).
On Sep. 12th (**Figure S3**), there were no significant vertical variations of isoprene and
monoterpenes across the altitudes. Notably, during the flight on Sep. 15th (**Figure S6**),
concentration enhancements can be seen at altitudes above 2000 m, suggesting a
potential contribution from atmospheric transport.

The ratio between VOC species is commonly applied to address the impact of
chemical removal and secondary formation during transport(Yang et al., 2024b;Zhu et
al., 2025). The vertical profile of the C8 aromatics-to-acetone concentration ratio was
plotted in **Figure 4** to demonstrate these effects. Both species can be emitted by

vehicular and industrial emissions (Jiang et al., 2023;Wang et al., 2024)), but C8 aromatics are more reactive, and acetone can be formed from secondary processes. The two effects both lead to a rapid decrease in the concentration ratio of C8 aromatics to acetone within the PBL before stabilizing as expected.

The averaged VOC concentrations measured below and above the PBL are compared in the scatter plot shown in **Figure 5**. To account for the variation of HPBL, data points above and below the light red area in **Figure 4** were used to calculate the averages and corresponding standard deviations. The averaged VOC concentrations below the PBL were consistently higher than those above the PBL, reflecting the combined effects of strong surface source emissions and the chemical oxidation process during vertical transport. Ethanol exhibited the highest concentration within the PBL with an average of 46.7 ppbv and the largest ratio of 16.5 between below- and above-PBL measurements. For the species with secondary formation, the data points of acetaldehyde, acetone, and MEK lay within the 2-5 ratio range. Above the PBL, the averaged concentrations of aromatic hydrocarbons were all smaller than 0.5 ppbv. Within the PBL, C8 aromatics showed the highest concentration, greater than the average above the PBL by a factor of 8.6, followed by toluene and benzene with factors of 6.6 and 5.4, respectively. C8 aromatics are quite chemically reactive, so a higher ratio suggests strong chemical removal, while for less reactive species, such as acetone, its ratio is closer to 1, indicating a weak impact of chemical reactions and potential contribution of secondary formation. Notably, the data points of styrene are clustered near the 1:1 ratio line, and occasionally, the concentrations above the PBL could be higher than those within the PBL. Since styrene is a primary species emitted at the surface, this pattern suggests a great contribution from transport, which will be discussed in **Section 3.3**. For isoprene, MVK and MACR, C9 aromatics, C10 aromatics, and acetonitrile, the average concentrations below the PBL were approximately twice as high as those measured above the PBL.

## 3.2 Differences in VOC vertical distribution in Beijing and Baoding

Two out of the six aerial surveys covered both the Beijing and Baoding areas. The aerial survey on Sep.14th, 2017 maintained a constant altitude of ~3000 m over Baoding, which prevented the collection of a vertical profile for Baoding. The flight conducted on Jul. 14th, 2019 provided comprehensive vertical distribution data for both cities through systematic altitude variations and therefore, was selected for the

comparative analysis of urban VOC profiles. As shown in **Figure S7**, the ascending
stage over Beijing was quite short (<10 mins), so the vertical profiles measured during
the descending stage at noon were selected to better represent the profiles in Beijing.
The spiral descending stage in Baoding was used in the comparative analysis as it had
well-designed altitude gradients.

The results of the comparison between the vertical VOC distributions of VOCs
in Beijing and Baoding are illustrated in **Figure 6**. The VOC profile over Beijing
exhibited notable concentration peaks around ~2500 m altitude, a feature previously
observed during the flight on Sep. 9[th], 2017 (**Figure S2)**. This anomaly will be discussed
in **Section 3.3** through the ratio analysis. For Baoding, almost all the VOCs showed
increasing trends with altitude, except for MVK and MACR, benzene, and toluene, with
stable low levels above 1000 m altitude throughout the observation periods.
Unfortunately, the aerial survey didn't capture the VOC surface measurements below
500 m in Baoding, which prevented the comparison between the near-ground VOC
levels in both cities. Based on the ground monitoring data from local air quality
monitoring stations (**Figure S8**), pollution episodes with ozone exceeding the Level-II
NAAQS in both Beijing and Baoding were captured on Jul. 14[th], 2019. Baoding
exhibited much higher ozone concentrations compared to Beijing, indicating more
severe photochemical pollution, and the northeast wind prevailing over Baoding
suggests potential influence from regional transport from Beijing, as reported in
previous studies (Huang et al., 2018). **Figure 7** presents the comparison between the
averaged vertical concentrations of VOCs in Beijing and Baoding during the same
altitude range (500-3000 m). Ethanol was found with the highest levels of VOCs
measured in the air above both cities (**Figure 7a**), 98 ppbv for Beijing and 153 ppbv
for Baoding, respectively. According to the scatter plot in **Figure 7b**, all the data points
cluster around the 1:2 line. The VOC concentrations in Baoding were higher than those
in Beijing by factors ranging from 1.2 to 3.5 with MEK showing the largest difference.
Much greater concentrations of C8 and C9 aromatics in Baoding were found than those
of benzene, which, together with the MEK showing the largest difference, suggests
more significant impacts from the industrial emissions on the air above Baoding. This
VOC enhancement implies a greater precursor reservoir over Baoding, which may
accelerate secondary pollutant formation through chemical oxidation production and
lead to more severe air pollution.

## 3.3 The contribution of regional emissions to VOC vertical profiles

Among six aerial surveys in 2017 and 2019, two distinct vertical distributions were noticed, characterized by abrupt VOC concentration enhancements above the PBL. For the aerial survey on Sep. 9[th], 2017, the vertical profile of styrene exhibited an inverse gradient, with concentrations increasing from 0.2-0.3 ppbv at ground to peak levels of 0.6-1.0 ppbv at ~3500 m altitude, as shown in **Figure 8a**. Above the ~3500 m altitude, styrene concentration rapidly fell back to levels close to the detection limit by 4000 m. Similar increases at 3500 m were also noticed for methanol, acetonitrile, acetaldehyde, acetone, benzene, and C9 aromatics. By plotting the ratios of averaged VOC levels at altitudes of 2500-3500 m to the ones near the ground (0-500 m) in **Figure 8b**, only styrene showed a ratio significantly larger than 1. This anomaly indicates that at altitudes of 2500-3500 m, the VOC levels were influenced by air mass rich in styrene transported from further regions.

We further conducted a correlation analysis with styrene and benzene, as they can be co-emitted from common anthropogenic sources. The paired data points of styrene and benzene are shown in **Figure 8d,** together with ratio ranges representing industrial emissions (Jiang et al., 2023;Zhong et al., 2017), vehicular emissions (Wang et al., 2022), and ratios measured in urban Beijing at the IAP tower. The data points were color-coded with altitude. An obvious transition between two subsets of data can be seen as the altitude increases. The data points near the ground are all lying between the lines representing the characteristic ratios of diesel vehicles (slope = 0.32) and gasoline vehicles (slope = 0.08) and are consistent with the ratio observed at the Beijing IAP tower. This indicates the dominant contribution from vehicular emissions for the VOCs near the ground. In contrast, the data points observed with altitudes between 2500-3500 m are located in the characteristic ratio ranges related to industrial emissions, which suggests that the VOCs within this altitude range were greatly impacted by industrial emissions. The synoptic chart in **Figure S9a** shows strong northerly winds at the 850 hPa level over the aircraft survey area, suggesting that the industrial emissions were likely transported from the north. Since styrene is more chemically reactive than benzene and thus the lifetime of styrene is much shorter than that of benzene. The ratio of styrene to benzene would decrease during transport. As shown in **Figure 8d**, the enhancement ratios at higher altitudes still fall within the characteristic ranges of

industrial sources and are significantly larger than those of vehicular emissions. Thus,
the chemical influences do not change our conclusion here.

During the aerial survey on Jul. 14[th], 2019, the VOC vertical profiles in the

Beijing area (**Figure 6**) showed similar trends with elevated VOC concentrations above
2500 m for almost all the VOCs. Analyses were conducted for this aerial survey as
shown in **Figure 9**. The vertical levels of acetonitrile (**Figure 9a**), a widely applied
tracer for biomass burning, were well above the typical backgrounds of 0.1-0.3 ppbv
reported in previous studies (Wu et al., 2016;Wang et al., 2007). The ratios of averaged
VOC levels at higher altitudes (2500-3000 m) and lower altitudes (<500 m) were
plotted in **Figure 9b**. Unlike the ratios illustrated in **Figure 8**, most VOC species in the
aerial survey on Jul. 14[th] had ratios around 1 or even greater than 2 for MEK, C10
aromatics, C9 aromatics, and styrene, showing the impact of significant regional
transport. Elevated acetonitrile concentrations above the background might suggest
biomass burning contribution. However, using acetonitrile as the biomass burning
tracer in urban regions can be problematic (Huangfu et al., 2021;Coggon et al., 2016).
Other sources (e.g., vehicular emissions) also emit acetonitrile, potentially interfering
with the identification of dominant emission sources (Inomata et al., 2013;Valach et al.,
2014). Further correlation analysis was conducted between acetonitrile and benzene in
**Figure 9d**. Both are typical pollutants emitted from biomass burning and vehicular
emissions, therefore, the specific emission ratio of acetonitrile and benzene for both
sources can be used as references in the source analysis. We applied the ratio measured
in central Beijing to represent the typical ratio of vehicular emissions. While only a few
data points are near the typical ratio of urban vehicular emissions (slope = 0.75), most
of the data points lie within the typical biomass burning ratio range, suggesting the
influence of biomass burning emissions. No clear height-dependent classification can
be found, which is distinguished from the analysis in **Figure 8d**. As shown in **Figure
S9b**, winds were quite weak at 850 hPa in the aircraft survey area, suggesting that the
elevated VOC concentrations were likely attributed to localized biomass burning
emissions.

The analysis of two distinct vertical distributions reveals that regional transport

of emissions, such as industrial and biomass burning, can significantly influence the
VOC vertical profiles, and the associated chemical processes and environmental impact
deserve further investigation.

## 4 Conclusion

Aircraft-based observations were conducted to investigate the vertical distribution of VOCs over Beijing and Baoding, two core cities in the NCP region. According to the vertical profiles of VOC concentrations, near-surface VOC levels were generally greater than those at higher altitudes, reflecting strong surface source emissions and chemical removal. In Beijing, ethanol exhibited the highest concentration within the PBL with an average of 46.7 ppbv and the largest vertical gradient with a ratio of 16.5 between the below- and above-PBL averages.

The vertical profiles over Beijing and Baoding were compared. Vertical-averaged VOCs above Baoding were generally greater than those in Beijing by factors ranging from 1.2 to 3.5. Increasing trends of concentrations with altitude in Baoding were observed for most VOC species, excluding MVK and MACR, benzene, and toluene. This implies richer precursors available for the secondary pollutant formation above Baoding.

Unlike the general vertical distributions, increases of VOC concentrations above 2500 m altitude were captured in Beijing. According to the ratio analysis, VOC levels near the surface were mainly emitted from vehicular emissions or under the joint impact of vehicular and biomass burning emissions. In contrast, the regional transport of industrial and biomass burning emissions drove the distinct enhancement of VOC concentration above the PBL.

This study presents vertical profiles of key VOC species up to ~4000 m in the core area of the NCP region, yielding valuable insights into VOC distribution patterns within the PBL and in the atmosphere above it. The observed concentration enhancements underscore the substantial impact of regional transport in shaping vertical distributions of VOCs. As these VOCs actively engage in the complex chemical processes above the PBL, the secondary pollutant formation at the higher altitudes necessitates further investigation.

## Data availability.

The data in this article are available from the corresponding author upon reasonable request.

**Supplement.**

The supplement related to this article is available online

**Author contributions.**

BY and JS designed the research. JS organized the aerial surveys. WZ, FW, PT, WX, YD, and JS contributed to data collection. YH and ZL performed the data analysis, with contributions from BY, SW, and XH. YH, ZL, and BY prepared the article with contributions from SW, XH, and JS. All the authors reviewed the article.

**Competing interests.**

The authors declare that they have no known competing financial interests that could have appeared to influence the work reported in this paper.

**Financial support.**

This work was supported by the National Key R&D Program of China (Grant No. 2022YFC3700604, 2023YFC3710900, 2023YFC3706103, 2024YFC3013001), the National Natural Science Foundation of China (Grant No. 42575211, 42230701, 42121004, 42275188), Guangdong Basic and Applied Basic Research Foundation (Grant No. 2025A1515011156), and the Innovation Foundation of CPML/CMA (2024CPML-C05). This work was also supported by Guangdong Provincial General Colleges and Universities Innovation Team Project (Natural Science) (Grant No. 2024KCXTD004) and Guangdong Province Special Support Plan for High-Level Talents (Grant No. 2023JC07L057).

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

**Table 1.** Detailed information for different flight routes during different aerial surveys in Sep. 2017 and Jul. 2019.

| Route number | Main area | Main altitude (m) | Date | Profile time* | |
|---|---|---|---|---|---|
| | | | | Ascending stage | Descending stage |
| Route 1 | Beijing | ~3800 | Sep. 09, 2017 | 12:06-12:29 | 12:31-16:54 |
| Route 2 | Beijing | ~3800 | Sep. 12, 2017 | 12:16-12:46 | 12:55-16:10 |
| Route 3 | Beijing | ~3800 | Sep. 13, 2017 | 13:35-14:11 | 14:30-16:55 |
| Route 4 | Beijing | ~3700 | Sep. 15, 2017 | 10:36-11:00 | 11:00-13:05 |
| Route 5 | Beijing | ~2500/~2800 | Sep. 14, 2017 | 12:31-12:46 | 16:00-16:58 |
| | Baoding | 2200-3500 | Sep. 14, 2017 | - | - |
| Route 6 | Beijing | ~3100 | Jul. 14, 2019 | 9:41-10:14 | 11:43-12:18 |
| | Baoding | ~3090 | Jul. 14, 2019 | 11:21-11:28 | 10:18-11:17 |

Note: * Profile time is the local time.

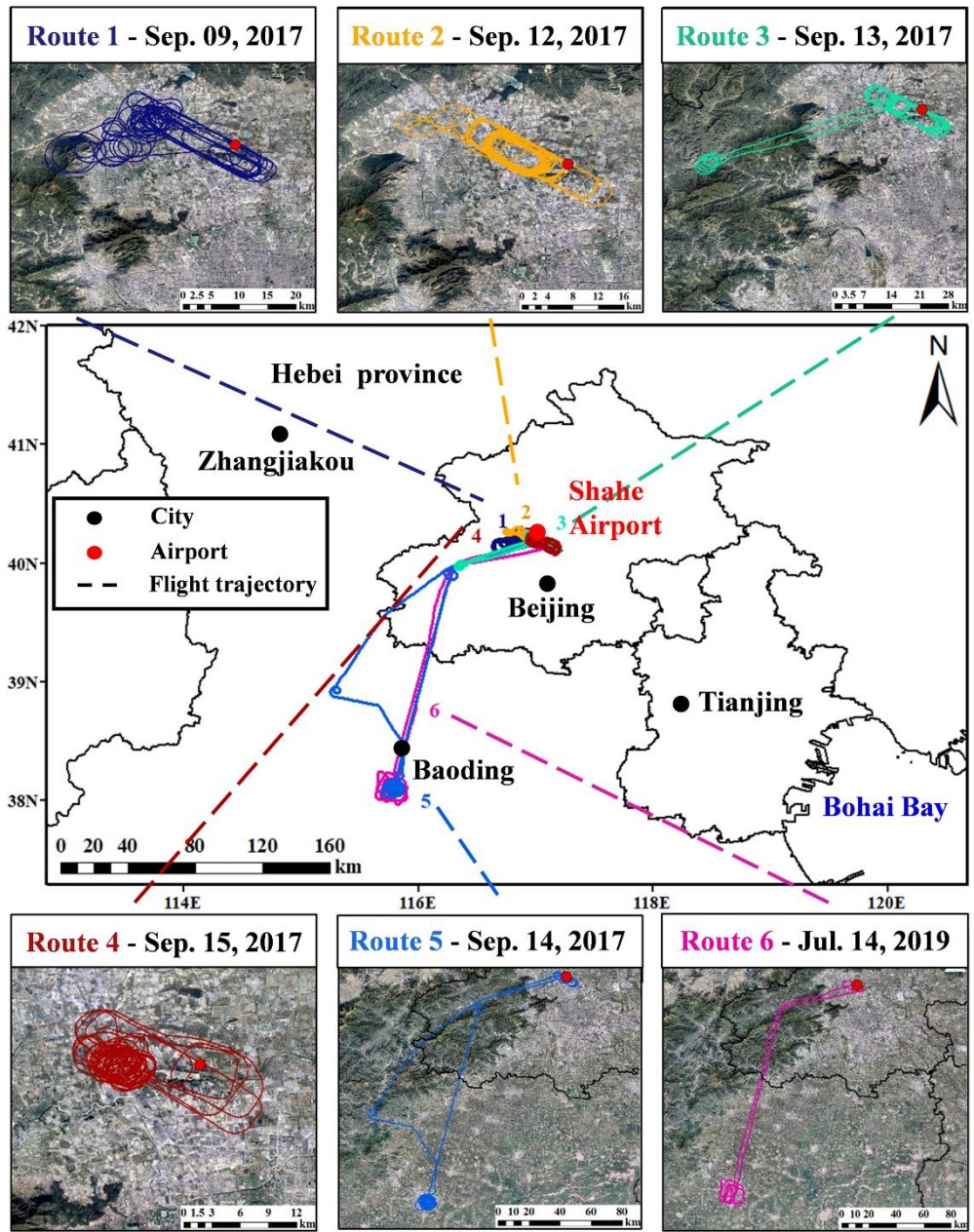


**Figure 1.** Flight trajectories during different aerial surveys in Sep. 2017 and Jul. 2019.
Figures were made by MeteoInfo software. The satellite images were downloaded from
©Google Earth and edited in ArcGIS 10.8.



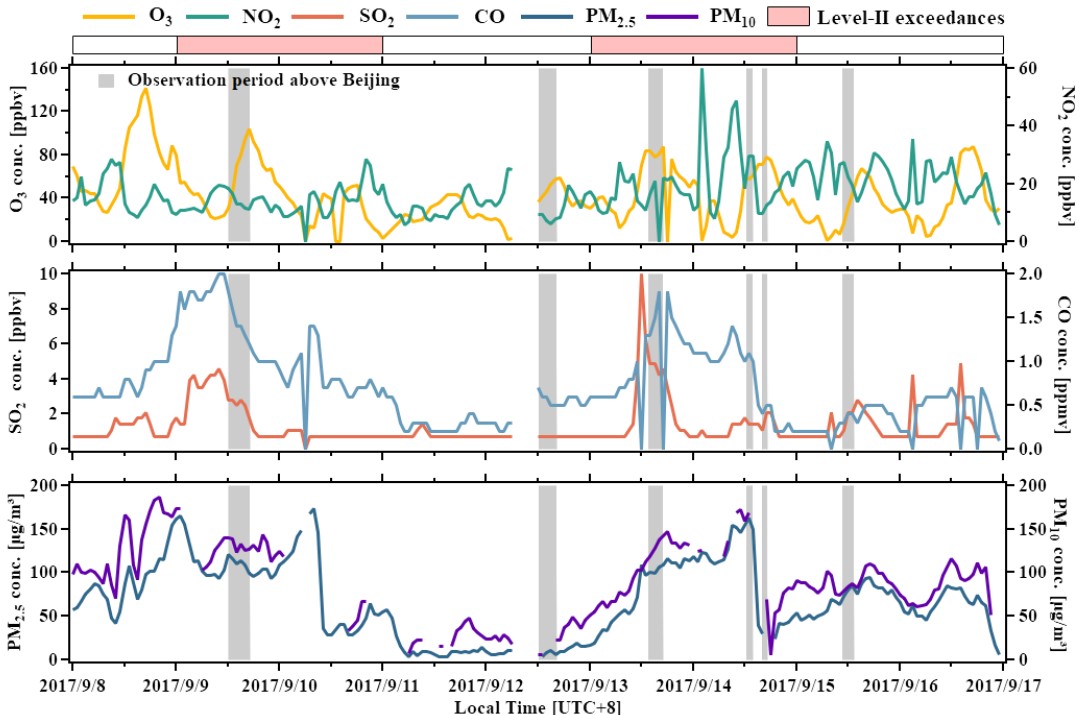

**Figure 2.** Time series of criteria pollutants including ozone, $NO_2$, $SO_2$, CO, $PM_{2.5,}$ and $PM_{10}$ from Sep. 8th to 16th, 2017. Data is obtained from Changping Town stations, the closest national air quality monitoring stations to the airport. Grey shaded areas indicate the observation periods. The bars filled with light red at the bottom show the periods when Level-II National Ambient Air Quality Standards (NAAQS) were exceeded.

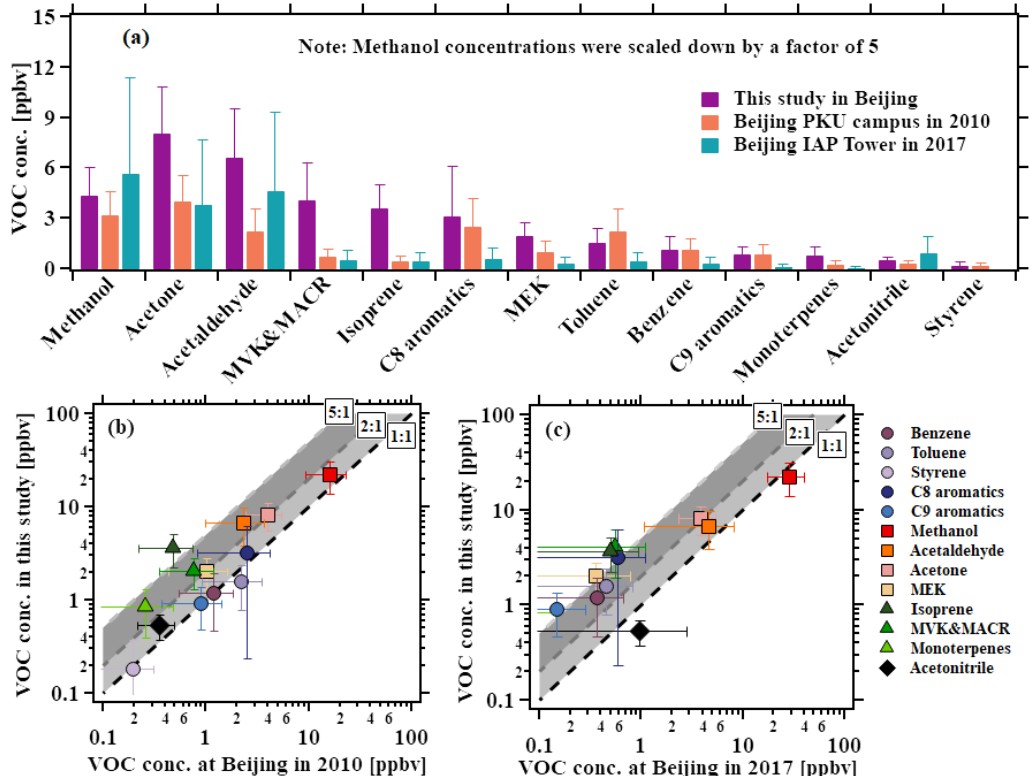

724

**Figure 3**. Comparison of averaged VOC concentrations below the PBL in this study and those measured in Beijing in 2010 (Yuan et al., 2012) and 2017 (Squires et al., 2020). Methanol concentrations were scaled down by a factor of 5 to improve visualization in (a). Scatter plots were also shown in (b) and (c). Error bars indicate the standard deviations. Reference lines are shown with shading to illustrate the differences.

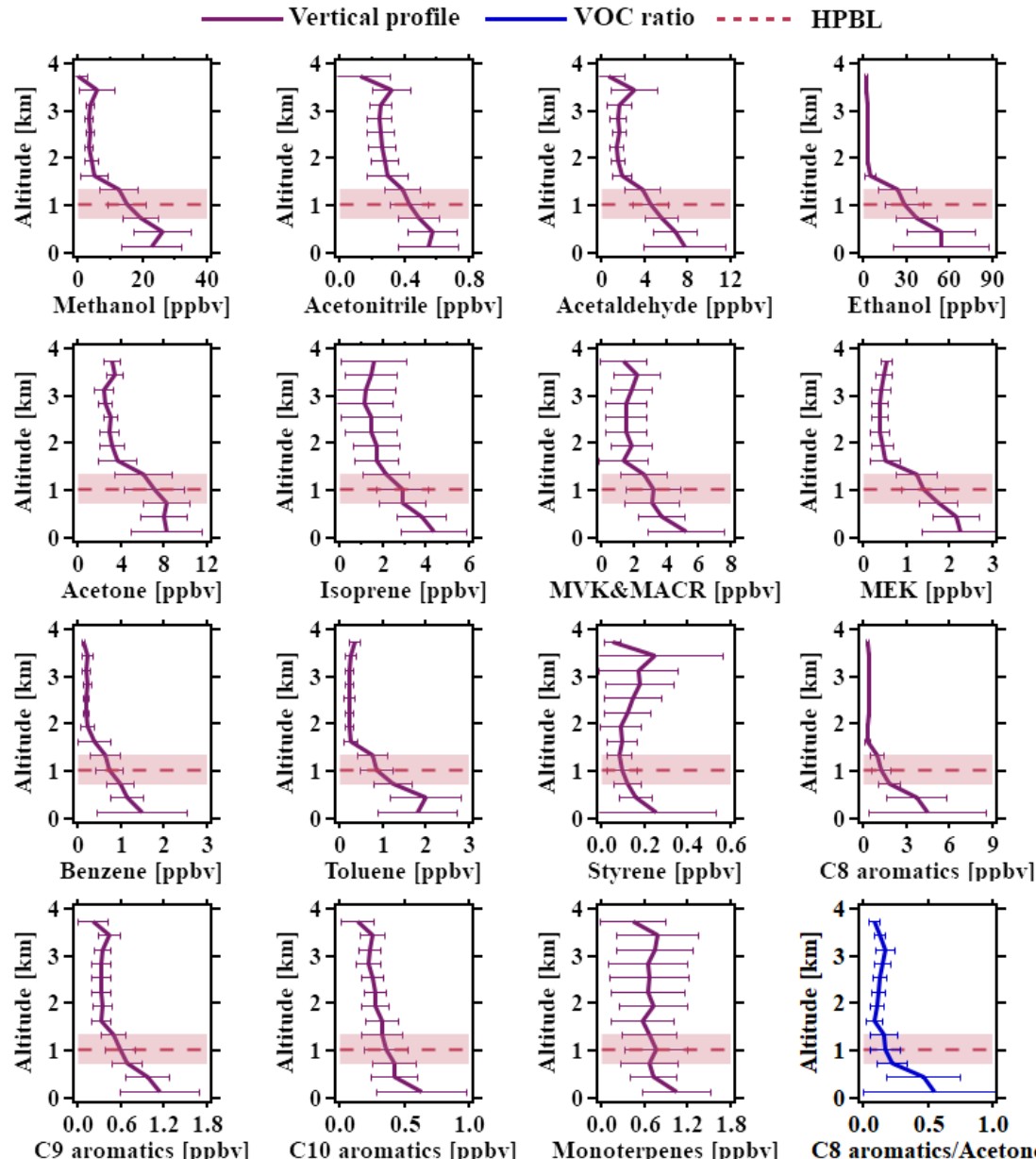


**Figure 4.** Averaged vertical profile (purple line) of VOCs in five aerial surveys above
the Beijing area in Sep. 2017 with error bar. The blue line shows the average vertical
profile of C8 aromatics-to-acetone concentration ratio with error bar. The red dashed
line is the average of the HPBL, with the light red area showing the variation range of
one standard deviation.

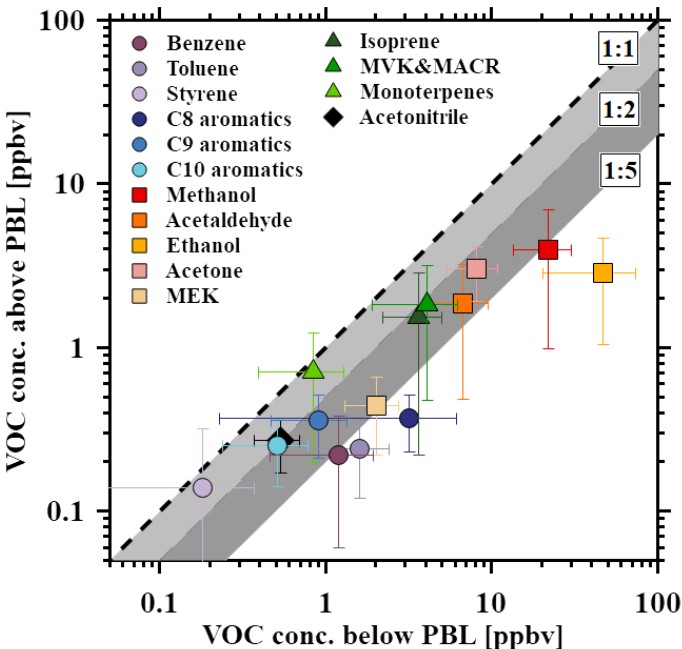


**Figure 5.** Scatter plot of the averaged VOC concentrations in Beijing below and above
the PBL in Sep. 2017. Error bars indicate the standard deviations. Reference lines are
shown with shading to illustrate the differences.

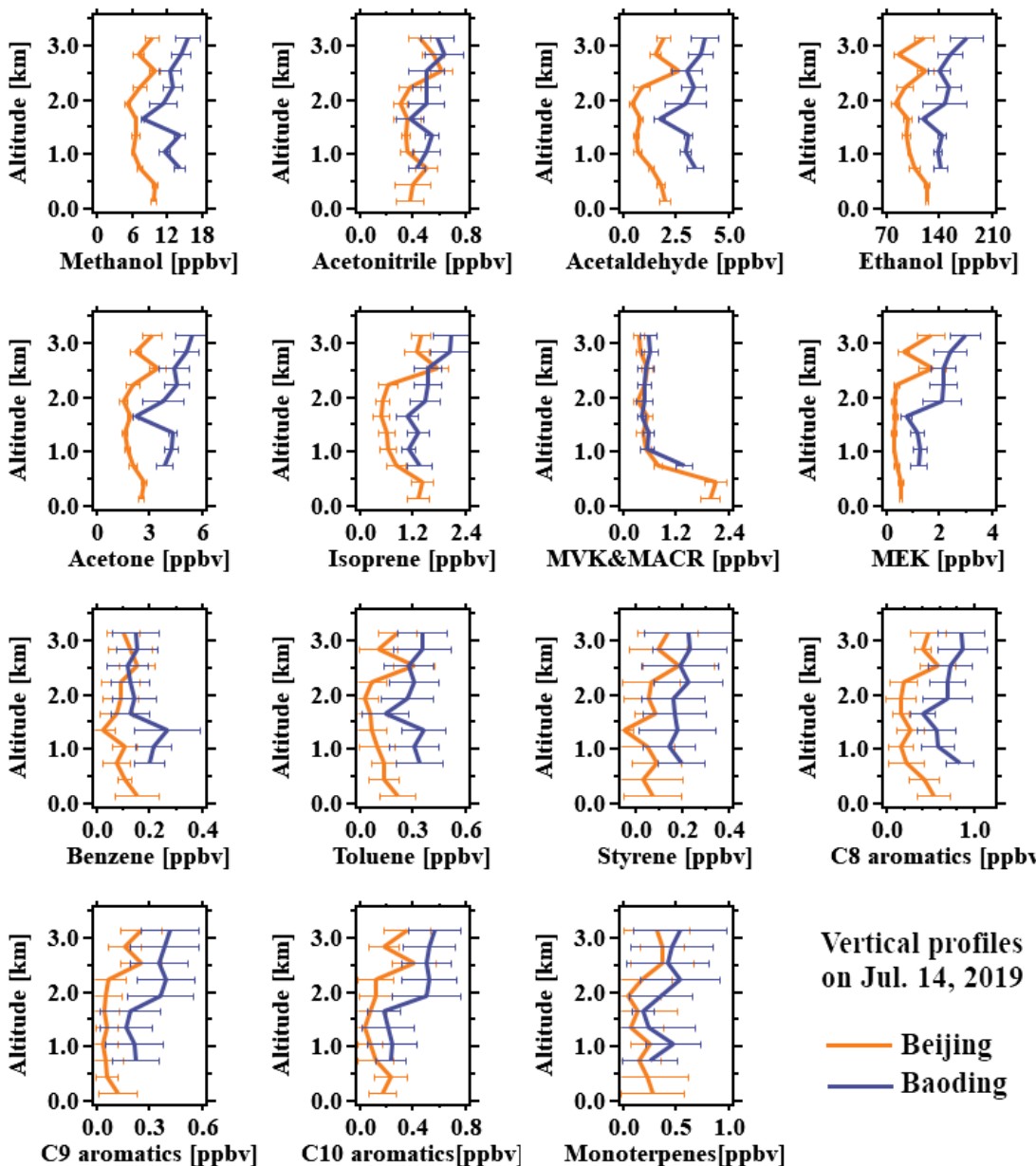


**Figure 6.** Comparison of vertical profiles of VOCs in Beijing (in orange) and Baoding

(in blue) during the aerial survey on Jul. 14[th], 2019, with error bars. The data measured

during the descending stages above both cities were plotted.

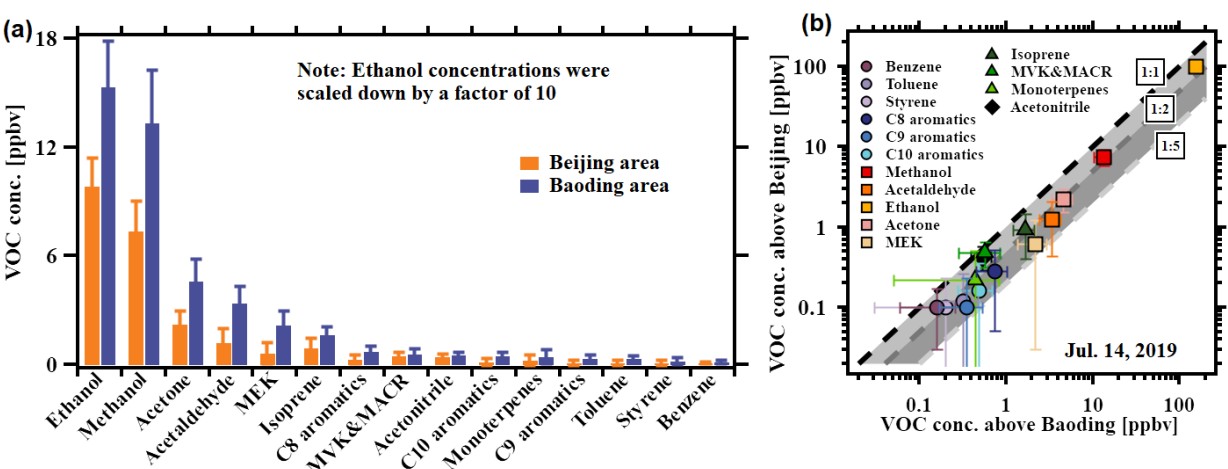

Figure 7. Comparison of averaged VOC vertical concentrations in Beijing and Baoding during the aerial survey on Jul. 14th, 2019. The data measured within the same altitude range (500 m - 3000 m) were averaged. Ethanol concentrations were scaled down by a factor of 10 to improve visualization in (a). The scatter plot was made in (b). Error bars indicate the standard deviations. Reference lines are shown with shading to illustrate the differences.

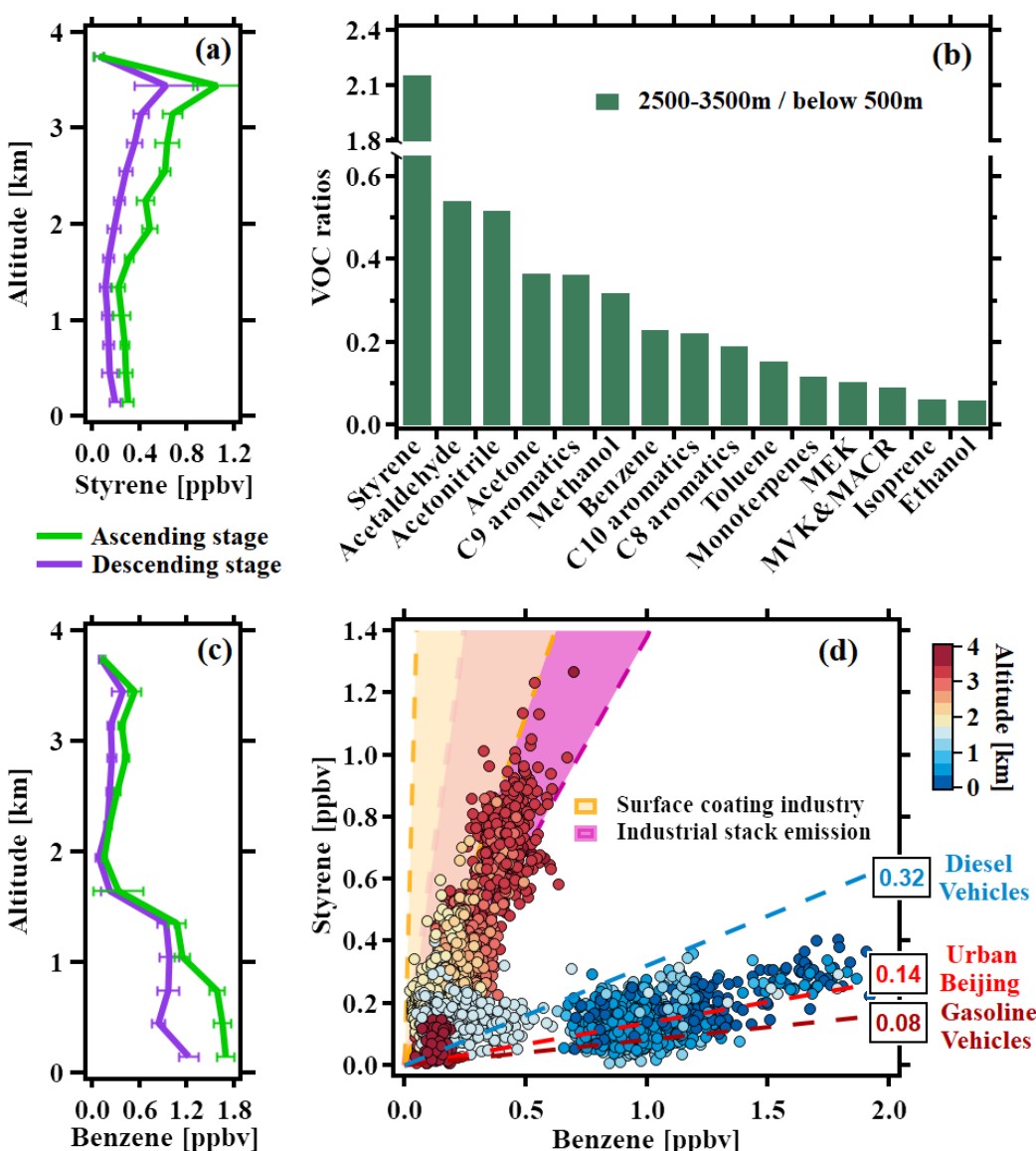

752

**Figure 8.** Analysis of the vertical profiles of styrene (a) and benzene (c) during ascending (in green) and descending (in purple) stages with error bars on Sep. 9th, 2017. (b) The ratios of each VOC species measured between 2500-3500 m and below 500 m. (d) The correlation analysis between styrene and benzene using the profiles. All the data points were color-coded with altitudes. Areas shaded with orange and purple represent the typical ratio ranges of styrene and benzene for the surface coating industry (Zhong et al., 2017) and industrial stack emissions (Jiang et al., 2023), respectively, showing an overlap between the two. The blue and dark red dashed lines represent ratios of styrene and benzene for diesel and gasoline vehicular emissions, respectively (Wang et al., 2024). The red dashed line represents the ratio measured in urban Beijing at the IAP tower in 2021.

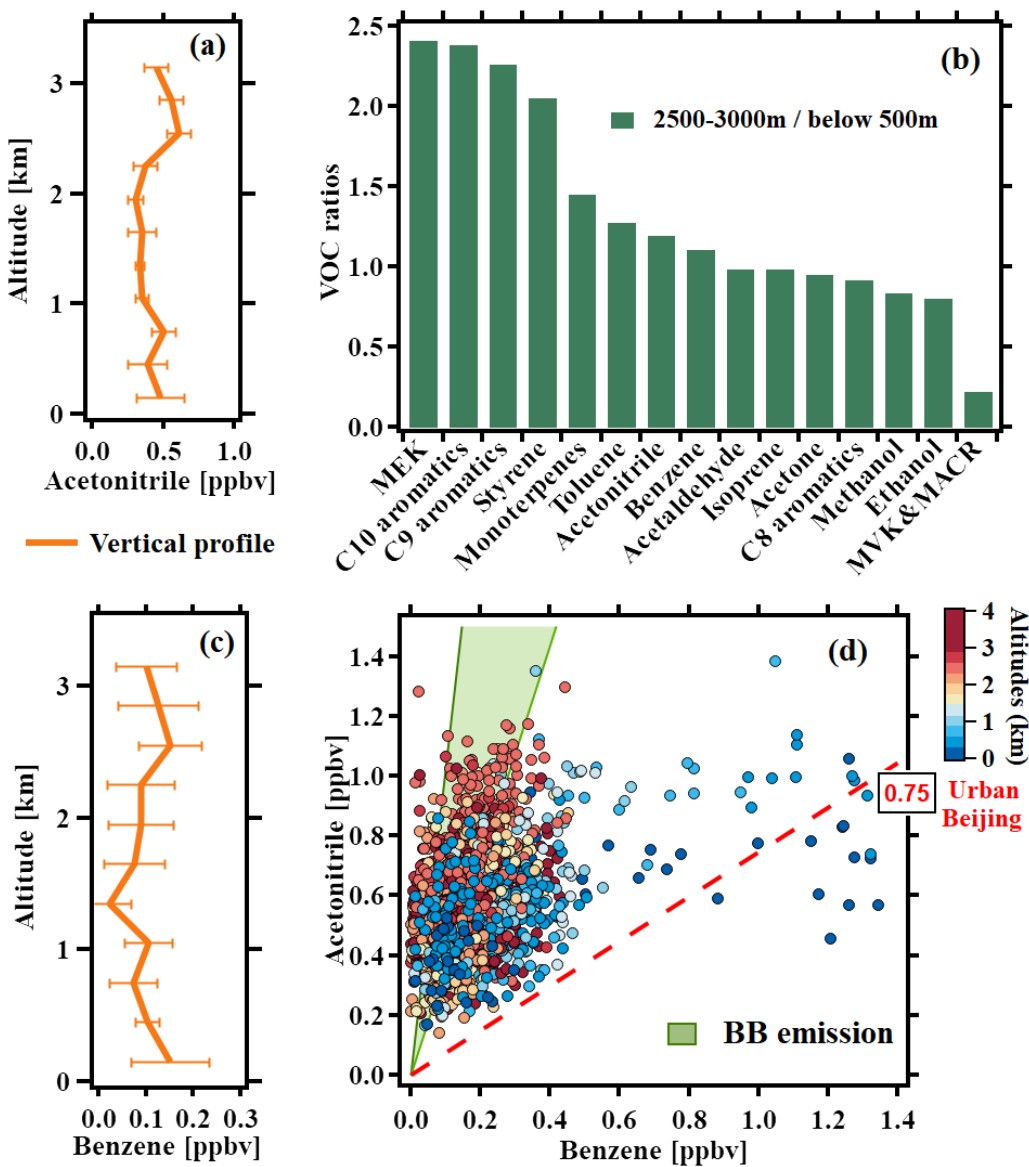

764

**Figure 9.** Analysis of the vertical profiles of acetonitrile (a) and benzene (c) with error
bars on Jul. 14th, 2019. (b) The ratios of each VOC species measured between 2500-
3000 m and below 500 m. (d) The correlation analysis between acetonitrile and benzene
using the whole flight data. All the data points were color-coded with altitudes. The red
dashed lines represent ratios in urban Beijing at the IAP tower in 2021. The area with
green shadow represents the ratio ranges of acetonitrile and benzene measured in the
biomass burning (BB) emissions, including wood, corncob, corn straw, and bean straw
(Gao et al., 2023).