# Peer review of "Aircraft-based observation of volatile organic compounds"

_EGUsphere, 2025_

## Author Comment (AC1)

**Response to Reviewer #1**

Huangfu et al. present airborne measurements over Beijing and Baodong to investigate the vertical distributions of VOCs above these cities. They identified that VOC concentrations were typically higher near the surface, with some exceptions during some flights, informing the sources, removal, and transport of these VOCs. Higher VOC concentrations were observed above Baoding across the whole vertical profile, pointing toward different regional emissions and photochemical processing. They utilize VOC ratios to determine the emission sources contributing to elevated VOC concentrations at higher altitudes during some flights, suggesting elevated industrial and biomass burning VOCs. This manuscript provides information on the vertical and spatial distribution of VOCs over highly populated cities, informing sources, removal, vertical transport, and photochemical production. I recommend this manuscript for publication following edits in response to the following comments.

The manuscript can be improved by including more context and discussion throughout the text, as identified by several of the specific comments below.

Reply: We thank you for your positive feedback and valuable comments. Each comment has been addressed in detail below and we hope our replies are to your satisfaction.

**Specific Comments:**

Line 16 – Include September 2017 and July 2019 so readers know the time of year for these observations.

Reply: Thank you for the suggestion. The related sentence has been revised in Lines 15-18. Please see below:

"Focusing on the core area of the North China Plain, aircraft-based observations were conducted in September 2017 and July 2019 to reveal the vertical distributions of volatile organic compounds (VOCs) measured by high-time resolution mass spectrometry."

Line 35-39 – This sentence is somewhat long and may be difficult to read for some people. I suggest splitting into two statements.

Reply: Thank you for your comment. We have revised the sentences accordingly. Please see Lines 36-41 in the revised manuscript:

"Following the implementation of the Action Plan on the Prevention and Control of Air Pollution, the particulate matter concentrations in NCP have significantly declined primarily due to a sharp reduction in anthropogenic emissions. In contrast, ozone levels have not shown a similar downward trend (Chen et al., 2020;Li et al., 2019a;Lu et al., 2019;Lu et al., 2020)"

Line 73 – The authors should provide additional context regarding the previous airborne VOC measurements to further bolster the importance of their study. For the 3 listed studies, which cities were the VOCs collected above? What were the time/spatial/vertical resolutions Which VOCs? Relevant conclusions to your study?

Reply: We thank you for your comments. Additional descriptions of the previous airborne VOC measurements have been added in Lines 77-84. Please see below:

"A typical decrease of non-methane hydrocarbons with increasing height was reported over Northeast China (Xue et al., 2011), while for the NCP region, aircraft-based measurements have been conducted but only reported the vertical distributions of BTEX species (benzene, toluene, ethylbenzene, xylenes), showing a similar negative trend with height (Liu et al., 2013). Nevertheless, the vertical distribution of VOCs in the NCP region is still unclear due to the scarcity of offline samples, hindering the ability to accurately assess the impacts of local emissions and air mass transport on VOC levels."

Line 105 – Can you provide more information on the "sampling device"?

Reply: Thank you for your suggestion. We have revised the related sentences. Please see Lines 110-113 in the revised manuscript:

"Ambient air was drawn through a 1.5-m-long PTFE tube at a flow rate of 15 L/min using a pump. A sub-stream of this air was then subsampled by the PTR-ToF-MS at a flow rate of 100 mL/min through a PTFE membrane particle filter."

Line 134-135 - I suggest moving "The vertical profiles of meteorological factors are shown in Figure S1." Up to ~line 118 along with measurements of meteorological parameters. Then revise the statement here (line 134-135) to "The vertical profiles of potential temperature..."

Reply: We thank you for the comment. The sentence in Lines 134-135 in the original manuscript has been removed up to Lines 135-136 in the revised manuscript as suggested and the statement in Lines 147-148 has been revised accordingly. Please see below:

"The vertical profiles of potential temperature for each flight can be found in Figure S1."

Section 2.2 – I believe more VOC quantification details are necessary.

When / how often was the PTR-MS calibrated? Zeroed?

Which standards were used to quantify the species listed in Table S1? For example, were monoterpenes calibrated based on the sensitivity of one isomer? Was MVK&MAC calibrated using an MVK standard, MACR standard, or both? This info could be included in Table S1 or elsewhere in the SI.

Were any measurements near the limit of detection? If so, the LODs should also be listed in the SI. If not, a brief statement in the methods would be helpful.

Typical propagated uncertainties for reported measurements should be included. Depending on the VOC, humidity would account for up to a 10% contribution (line 115) on top of the measurement error and error propagation through the calibration.

Have you considered any interfering species in your measurements? For example, ethylbenzene impacting benzene signal, or aldehydes from cooking emissions impacting isoprene signal? See e.g., Coggan et al. (2024).

Reply: We thank you for your comments regarding the VOC quantification details. The PTR-ToF-MS instrument was calibrated before each field campaign. We used a commercial gas standard with 21 VOC species prepared by Apel Riemer Environmental Inc. Specifically, a-pinene, MVK, and MACR in the standard gas were used for calibrating the sensitivities of monoterpenes and MVK&MACR, respectively. Please see the information on standard gas below:

Table R1. VOC species (concentration and uncertainty) contained in the standard gas

| Compound     | CAS#     | Concentration (ppb) | Uncertainty |
|--------------|----------|---------------------|-------------|
| Formaldehyde | 50-00-0  | 1076                | ±5%         |
| Propane      | 74-98-6  | 1004                | ±5%         |
| 1-Butene     | 106-98-9 | 1000                | ±5%         |

| Acetaldehyde           | 75-07-0  | 1028 | ±5% |
|------------------------|----------|------|-----|
| Methanol               | 67-56-1  | 1018 | ±5% |
| Ethanol                | 64-17-5  | 1091 | ±5% |
| Isoprene               | 78-79-5  | 1000 | ±5% |
| Acetone                | 67-64-1  | 1026 | ±5% |
| Dimethyl sulfide       | 75-18-3  | 1009 | ±5% |
| Acetonitrile           | 75-05-8  | 1056 | ±5% |
| Methacrolein           | 78-85-3  | 1015 | ±5% |
| Methyl Vinyl Ketone    | 78-94-4  | 1013 | ±5% |
| Methyl Ethyl Ketone    | 78-93-3  | 1044 | ±5% |
| Benzene                | 75-18-3  | 1032 | ±5% |
| Toluene                | 108-88-3 | 1012 | ±5% |
| Ethylbenzene           | 100-41-4 | 1004 | ±5% |
| Styrene                | 100-42-5 | 997  | ±5% |
| a-Pinene               | 80-56-8  | 987  | ±5% |
| 1,3,5-Trimethylbenzene | 108-67-8 | 1004 | ±5% |
| Benzaldehyde           | 100-52-7 | 1070 | ±5% |
| 1,4-Diethylbenzene     | 105-05-5 | 1079 | ±5% |
|                        |          |      |     |

The limits of detection (LODs) for all VOCs reported in this study have been added to **Table S1**. The overall uncertainties were propagated and listed in **Table S1**, including the standard gas uncertainty (5%), dilution uncertainty (1%), a random error (5% - 33%) determined based on Poisson statistics, and the uncertainty caused by humidity (2% - 9.5%). The interferences of fragmentations were not corrected due to the lack of necessary VOCs data. Additional details on VOC quantification by PTR-ToF-MS have been added in the revised manuscript (Lines 118-132):

"The sensitivities of PTR-ToF-MS for various VOC species were calibrated with commercial standard gas (Apel-Riemer, Environmental Inc., USA) before each field campaign. The sensitivity of monoterpenes was calibrated based on the a-pinene in the standard gas. Both methyl vinyl ketone (MVK) and methacrolein (MACR) were included in the standard gas, so the sensitivity of MVK&MACR was calculated based on their summed concentrations. A total of 15 VOC species are reported in this study and listed in Table S1 as well as the limits of detection (LODs) and propagated uncertainties. Based on the multi-level tests in the laboratory, the impacts of humidity on VOC sensitivities were evaluated to be less than 10% for the reported VOC species, so no

correction was conducted, and the induced uncertainty was propagated to the overall uncertainties. The interferences of fragmentations, such as the impact of higher-carbon aldehydes and cycloalkanes on isoprene signal (m/z 69,  $C_5H_8H^+$ ) and the impact of ethylbenzene on benzene signal (m/z 79,  $C_6H_6H^+$ ), were not corrected, so the concentrations of isoprene and benzene might be overestimated due to these interferences."

Line 141 – Include the distance between the air quality station and the airport.

Reply: Thank you for your suggestion. The distance between the air quality station and the airport has been included in the related sentence. Please see Lines 155-158 in the revised manuscript:

"The ambient air quality data of criteria pollutants (ozone,  $NO_2$ ,  $SO_2$ , CO,  $PM_{2.5}$ , and  $PM_{10}$ ) of Changping Town station, the nearest national air quality monitoring station to the Shahe Airport (11 km away), were collected from the China National Environmental Monitoring Center (CNEMC) (**Figure 2**)"

Line 144 – Please include the full term for NAQS before the acronym. What are the Level II threshold values? It would be useful to add corresponding horizontal bars to Figs. 2 and S8 to show the reader when ground-level exceedances happened relative to the flights.

Reply: We thank you for your comments. The full term of the acronym has been added as suggested. Please see Lines 158-161 in the revised manuscript:

"According to the Level-II thresholds in China's current National Ambient Air Quality Standards (NAAQS), pollution events in Beijing with PM2.5 exceedances on Sep. 9th, 10th, and 14th and ozone exceedances on Sep. 13th were noticed."

Horizontal bars have been added to Figs. 2 to show the period corresponding to the exceedances and the caption has been revised accordingly. For the Sep. 2019 measurement, a series of continuous exceedances from Jul. 12th to 16th, 2019 was observed covering both Beijing and Baoding. Therefore, we have added a sentence to describe these exceedances in the figure caption without updating Figure S8:

**Figure 2.** Time series of criteria pollutants including ozone,  $NO_2$ ,  $SO_2$ , CO,  $PM_{2.5}$ , and  $PM_{10}$  from Sep.  $8^{th}$  to  $16^{th}$ , 2017. Data is obtained from Changping Town stations, the closest national air quality monitoring stations to the airport. Grey shaded areas indicate the observation periods. The bars filled with light red at the bottom show the periods when Level-II National Ambient Air Quality Standards (NAAQS) were exceeded.

Figure S8. Time series of criteria pollutants including ozone, NO2, SO2, CO, PM2.5, and PM10 from Jul. 12th to 16th, 2019 over Beijing (a) and Baoding (b). The Data is obtained from the Beijing Changping Town station and the Baoding natatorium station, the closest national air quality monitoring stations to the flight trajectories. Grey shaded areas indicate the observation periods. According to the Level-II National Ambient Air Quality Standards (NAAQS), continuous exceedances from Jul. 12th to 16th, 2019 were observed covering both Beijing and Baoding.

Line 152 – What do the errors represent? The standard deviation of all measurements across all September 2017 flights?

Reply: Thank you for your comment. The concentration average and standard deviation

of all 15 VOCs reported were calculated and listed in **Table S3** for each flight in Sep. 2017. The values in Line 152 describe the range of concentration averages of the five flights and the error represents the standard deviations along with the average for the specific flight. We have revised the sentences in Lines 167-169:

"Five flights were conducted over Beijing in Sep. 2017. The averaged concentrations of all 15 VOC species ranged from  $25.9 \pm 13.4$  ppb measured on Sep.12th, 2017 to 52.1  $\pm$  57.7 ppb measured on Sep.14th, 2017 (**Table S3**)"

Line 154 – Provide more context regarding the other studies for the reader. Include the time of year when the previous measurements were made, that they were ground-based, and the measurement techniques. For the 2010 study, are you comparing against the PTR data or the GC data, since that could skew your comparison?

Reply: We thank you for this comment. For the 2010 study, we compared our data against the PTR-MS data reported in Yuan et al., 2012. We have added additional descriptions of the other studies. Please see Lines 173-177 and Lines 184-187 in the revised manuscript:

"Yuan et al. (2012) conducted VOC measurements on the top of a six-story building on Peking University campus using the PTR-MS technique in the summer of 2010. The concentrations of aromatic species below the PBL in this study were comparable to those measured in 2010 (Figure 3b), with data points clustering along the 1:1 line."

"VOC measurements by PTR-ToF-MS in the summer of 2017 were conducted at the 102 m platform of the Institute of Atmospheric Physics (IAP) meteorology tower and represented the VOC concentrations driven by traffic-related emissions in the center of urban Beijing (Squires et al., 2020)."

Line 156 – Provide more context to explain why the 2010 measurements were comparable to your measurements. Later you compare to the 2017 IAP measurements and explain the differences based on measurement location and traffic/industry influence. Please do the same for this comparison to 2010. Were the 2010 measurements also primarily influenced by industry?

Reply: Thank you for your suggestions. The air quality in Beijing has been improving with VOC levels declining as a result of the implementation of stricter emissions

standards (Wang et al., 2015; Yao et al., 2022). Regarding the comparison in Figure 3, both the Peking University site and the IAP site are typical urban sites, while the aircraft surveys were conducted in suburban Beijing. The discrepancy in measuring locations may lead to greater impacts from industrial emissions on VOC levels measured in the aircraft surveys, as industrial emissions are mainly distributed in suburban areas (Wang et al., 2015). The Peking University measurement and aircraft surveys were conducted seven years apart, during which both traffic and industrial emissions in urban Beijing declined significantly. Therefore, we attribute the similarity in VOC levels between the two measurements in Figure 3b to be coincidental, due to the combined influences of larger emissions (both traffic and industrial) in urban Beijing in 2010 and the greater contribution of industrial emissions in suburban Beijing in 2017. We have added additional discussions in the revised manuscript (Lines 178-183):

"The two datasets in **Figure 3b** were measured seven years apart, during which VOC emissions in urban Beijing declined significantly (Wang et al., 2015; Yao et al., 2022a). While the VOC levels at the campus site in 2010 should have been higher due to greater VOC emissions, this effect was likely compensated for by the extra industrial emissions in suburban areas, which might be the reason for the comparable results observed." References:

Wang, M.; Shao, M.; Chen, W.; Lu, S.; Liu, Y.; Yuan, B.; Zhang, Q.; Zhang, Q.; Chang, C. C.; Wang, B.; et al. Trends of non-methane hydrocarbons (NMHC) emissions in Beijing during 2002–2013. Atmospheric Chemistry and Physics 2015, 15 (3), 1489-1502. DOI: 10.5194/acp-15-1489-2015.

Yao, D.; Tang, G.; Sun, J.; Wang, Y.; Yang, Y.; Wang, Y.; Liu, B.; He, H.; Wang, Y. Annual nonmethane hydrocarbon trends in Beijing from 2000 to 2019. Journal of Environmental Sciences 2022, 112, 210-217. DOI: 10.1016/j.jes.2021.04.017.

**Line 160 – Provide full term for IAP before acronym.**

Reply: Thank you for this suggestion. The full term of IAP has been added in Lines 184-187. Please see below:

"VOC measurements by PTR-ToF-MS in the summer of 2017 were conducted at the 102 m platform of the Institute of Atmospheric Physics (IAP) meteorology tower and

represented the VOC concentrations driven by traffic-related emissions in the center of urban Beijing (Squires et al., 2020)."

Line 160 – Add "and": "... meteorology tower and represented..."

Reply: We thank you for the comment. The related sentence has been revised in the revised manuscript (Lines 184-187):

"VOC measurements by PTR-ToF-MS in the summer of 2017 were conducted at the 102 m platform of the Institute of Atmospheric Physics (IAP) meteorology tower and represented the VOC concentrations driven by traffic-related emissions in the center of urban Beijing (Squires et al., 2020)."

Line 163 – Regarding traffic vs industrial emissions when comparing to the 2017 IAP measurements: Fig. 3c suggests a greater enhancement in the C8 and C9 aromatics compared to benzene and toluene. You should discuss this observation in the context of the literature, comparing distributions of the aromatics you observed against typical traffic emissions and industrial solvents (as you do with MEK later, line 137).

Reply: We thank you for this suggestion. We have added additional discussions to address the greater enhancement of C8 and C9 aromatics. Please see Lines 189-194 in the revised manuscript:

"The enhancements of C8 and C9 aromatics are much greater than those of benzene and toluene. Given that industrial emissions contain higher proportions of C8 and C9 aromatics (Wang et al., 2024; Jiang et al., 2023), these results suggest that the VOC measured in aerial surveys in this study might be under the impacts of industrial emissions from the suburban region, especially at lower altitudes."

Line 167 - Perhaps industrial tetrahydrofuran could also contribute to "MEK" (C4H8OH+)?

Reply: Thank you for your comment. We agree that tetrahydrofuran, as an isomer of MEK, is also an important industrial pollutant. We have revised the relevant sentence accordingly. Please see Lines 196-201 in the revised manuscript:

"As one of the common ingredients in industrial solvents, MEK can be emitted through multiple industrial processes (Wu et al., 2020b; Wang et al., 2024). Similarly, tetrahydrofuran (THF), an isomer of MEK and a significant industrial pollutant itself

(Hu et al., 2018), may also contribute to the measured MEK signals. Hence, the higher concentrations of both compounds in this study are likely attributable to the nearby industrial emissions."

Line 174 – Somewhere, it would be useful to briefly discuss your vertical distributions in the context of the other airborne VOC studies you cited in the introduction (lines 73-76). Even though these studies are in different cities, do the vertical profiles generally agree? Why or why not?

Reply: Thank you for your suggestion. We have added additional discussions about the comparison between our vertical distributions and the ones reported in the literature. Please see Lines 212-215 in the revised manuscript:

"These general decreasing trends of pollutants have been reported by previous studies (Benish et al., 2020;Liu et al., 2013;Xue et al., 2011). However, due to a lack of high-resolution VOC measurements, concentration variations with height and anomalous enhancement have not been documented."

Line 191 – For the composite profiles, the increase in styrene (and the OVOCs) with altitude appears to be driven by one flight (Fig. S2). Currently the discussion implies that styrene, acetonitrile, and the OVOCs were increasing with altitude for all flights. Add a clear explanation about the impact of the one flight on the composite profile.

Reply: Thank you for your comment. We have added one sentence to clarify the impact of one flight on the composite profile in Lines 228-230. Please see below:

"By checking the profiles of each flight (**Figures S2-S6**), such enhancements in the composite profiles were mainly driven by the measurement on Sep. 9th, 2017."

Line 194 – Following from my previous comment, it would be useful to distinguish that the analysis of biomass burning and industrial sources in Section 3.3 refers to different profiles. Following my initial reading, I assumed the discussion would be in reference to a single profile.

Reply: Thank you for your suggestion. We have revised the related sentences. Please see Lines 228-233 in the revised manuscript:

"By checking the profiles of each flight (**Figures S2-S6**), such enhancements in the composite profiles were mainly driven by the measurement on Sep. 9th, 2017. This

anomalous profile, potentially associated with long-range transported industrial emissions, will be further explored in **Section 3.3**. Similar VOC concentration enhancements were also found on the flight for Jul. 14th, 2019, which will be analyzed as well in **Section 3.3**."

Line 226 – I'm not sure what you mean by grey area in Fig. 4. Are you referring to the red shaded areas?

Reply: We appreciate you pointing out this lack of clarity in the discussion about Fig.4. The related sentence has been revised in Lines 271-273. Please see below:

"To account for the variation of HPBL, data points above and below the light red area in **Figure 4** were used to calculate the averages and corresponding standard deviations."

Lines 233-237 – Break this sentence into two parts. The "above the PBL" and "within the PBL" portions are two distinct points.

Reply: Thank you for your comment. We have revised the sentences in Lines 279-283 accordingly. Please see below:

"Above the PBL, the averaged concentrations of aromatic hydrocarbons were all smaller than 0.5 ppbv. Within the PBL, C8 aromatics showed the highest concentration, greater than the average above the PBL by a factor of 8.6, followed by toluene and benzene with factors of 6.6 and 5.4, respectively."

Line 268 – What was the predominant wind direction? Was Baoding downwind of Beijing or other cities, contributing to an accumulation of pollutants and the formation of photochemical products during transport?

Reply: We thank you for this comment. During the flight on Jul. 14th, 2019, the predominant wind direction at Baoding is northeast, suggesting the air above Baoding could be under the impact of regional transport from Beijing. Previous research also found that the background and regional transport contributed about half of the concentration of surface ozone at Baoding (Huang et al., 2018). We have added additional discussions in the revised manuscript (Lines 315-319):

"Baoding exhibited much higher ozone concentrations compared to Beijing, indicating more severe photochemical pollution, and the northeast wind prevailing over Baoding suggests potential influence from regional transport from Beijing, as reported in previous studies (Huang et al., 2018)"

**Reference:**

Huang Z., Hong L., Yin P., Wang X., Zhang Y. Source Apportionment and Transport Characteristics of Ozone in Baoding during Summer Time[J]. Acta Scientiarum Naturalium Universitatis Pekinensis, 2018, 54(3): 665-672.

Line 274 – Provide some discussion as to which VOCs had the higher and lower ratios between the cities and why. For example, C9 aromatics appear to be enhanced by a factor of ~3 while benzene is ~2 (Fig. 7). Differences in these aromatics suggest either (1) different emission sources dominate the aromatics between the cities, or (2) there is a greater degree of oxidation for the aromatic emissions in Beijing compared to Baodong.

Reply: We thank you for this comment. We have added additional discussions about the VOC ratios between cities. Please see Lines 324-328 in the revised manuscript:

"The VOC concentrations in Baoding were higher than those in Beijing by factors ranging from 1.2 to 3.5 with MEK showing the largest difference. Much greater concentrations of C8 and C9 aromatics in Baoding were found than those of benzene, which, together with the MEK showing the largest difference, suggests more significant impacts from the industrial emissions on the air above Baoding."

Lines 285-286 – It appears that acetone was also slightly enhanced at altitude. Is there a reason it was excluded in the text?

Reply: Thank you for this comment. The acetone concentration did increase at ~3500 m and the related sentence has been revised in Lines 339-340. Please see below:

"Similar increases at 3500 m were also noticed for methanol, acetonitrile, acetaldehyde, acetone, benzene, and C9 aromatics."

Line 304 – The ratio analysis compares against fresh emissions. You should comment on the lifetime of styrene vs benzene in the context of transport times from the surface, and how that would affect your observed ratios.

Reply: We thank you for your comment. Styrene is more chemically reactive than benzene and thus the lifetime of styrene is much shorter than that of benzene. The ratio of styrene to benzene would decrease during transport. As shown in Fig. 8d, the

enhancement ratios at higher altitudes still fall within the characteristic ranges of industrial sources and are significantly larger than those of vehicular emissions. Therefore, our conclusion remains robust. We have added additional discussions in the revised manuscript (Lines 360-365):

"Since styrene is more chemically reactive than benzene and thus the lifetime of styrene is much shorter than that of benzene. The ratio of styrene to benzene would decrease during transport. As shown in Figure 8d, the enhancement ratios at higher altitudes still fall within the characteristic ranges of industrial sources and are significantly larger than those of vehicular emissions. Thus, the chemical influences do not change our conclusion here".

Line 321 – Acetonitrile and benzene don't appear to correlate well in Fig. 9d, suggesting different sources. Please include some discussion. Did acetonitrile correlate better with a different VOC which could better support the biomass burning (or other) source? Reply: We thank you for your comment. In the original manuscript, Fig. 9d was plotted using the profile data shown in Fig. 9a and 9c. We have updated Fig. 9d using the data measured in the whole flight measurement on Jul. 14th, 2019 and a better correlation can be revealed, likely suggesting the impact of biomass burning emissions. Please see below:

Figure 9. Analysis of the vertical profiles of acetonitrile (a) and benzene (c) with error bars on Jul. 14th, 2019. (b) The ratios of each VOC species measured between 2500-3000 m and below 500 m. (d) The correlation analysis between acetonitrile and benzene using the whole flight data. All the data points were color-coded with altitudes. The red dashed lines represent ratios in urban Beijing at the IAP tower in 2021 (He et al., 2025). The area with green shadow represents the ratio ranges of acetonitrile and benzene measured in the biomass burning (BB) emissions, including wood, corncob, corn straw, and bean straw (Gao et al., 2023).

We have revised the related discussion accordingly. Please see Lines 385-388 in the revised manuscript:

"While only a few data points are near the typical ratio of urban vehicular emissions (slope = 0.75), most of the data points lie within the typical biomass burning ratio range, suggesting the influence of biomass burning emissions."

Table 1 – It looks like most columns were partially cut off (e.g., column 2 lists "Beijin", the date column lists Sep. 09, "201"). It's not clear to me if the error is on my end, but please double check.

Reply: Thank you for your comment. We have double-checked Table 1 and adjusted the format for a better vision.

Tables 1 and S2 – I assume the times are local? Please clarify.

Reply: Thank you for pointing out this lack of clarity in Tables 1 and S2. Yes, all the times in Tables 1 and S2 are local times. Like Table 1, we have added a note below Table S2 to clarify.

Figure 8 – The caption and legend mention two types of industrial emissions and the corresponding shaded areas. I see three shaded areas (light orange, dark orange, purple). Does the dark orange region in the middle represent an overlap between the two types of industrial emissions? Please clarify in the caption.

Reply: We appreciate you pointing out this color issue in the figure caption. We have revised the caption of Fig. 8. Please see Lines 751-761 in the revised manuscript: "Analysis of the vertical profiles of styrene (a) and benzene (c) during ascending (in green) and descending (in purple) stages with error bars on Sep. 9th, 2017. (b) The ratios of each VOC species measured between 2500-3500 m and below 500 m. (d) The correlation analysis between styrene and benzene using the profiles. All the data points were color-coded with altitudes. Areas shaded with orange and purple represent the typical ratio ranges of styrene and benzene for the surface coating industry (Zhong et al., 2017) and industrial stack emissions (Jiang et al., 2023), respectively, showing an overlap between the two. The blue and dark red dashed lines represent ratios of styrene and benzene for diesel and gasoline vehicular emissions, respectively (Wang et al., 2024). The red dashed line represents the ratio measured in urban Beijing at the IAP

Table S1 – Remove the extra "c" in "C8 acromatics"

tower in 2021 (He et al., 2025)"

Reply: Thank you for pointing out this typo. The extra "c" in C8 aromatics has been removed.

Table S2 – I suggest adding to the caption "as discussed in Section 2.3" or similar. I also suggest reiterating the 10% uncertainty (mentioned on line 136).

Reply: Thank you for your suggestions. We have revised the caption of Table S2 accordingly. Please see below:

"Table S2. The heights of the planetary boundary layer (HPBL) of Beijing determined by the air parcel method during all the aerial surveys in Sep. 2017 and Jul. 2019 as discussed in Section 2.3. A 10% uncertainty is assigned to the results."

Table S3 – Is there a purpose for the superscript "c" above MEK?

Reply: Thank you for pointing out this typo. The superscript "c" above MEK has been removed from Table S3.

Tables S3 and S4 – You provide summary statistics for overall observations (Table S3) and below the PBL (Table S4). I suggest adding another SI table for above the PBL since you use those values in Fig. 5.

Reply: We thank you for your suggestion. Table S5 has been added with the VOC concentrations above the PBL. We have added one sentence in Lines 169-170 in the revised manuscript:

"The VOC concentrations measured within and above the PBL are listed in Tables S4 and S5."

---

## Author Comment (AC2)

**Response to Reviewer #2**

In this manuscript, the authors present aircraft-based vertical observations of volatile organic compounds (VOCs) measured using high-time resolution mass spectrometry. Despite the technical challenges associated with such data collection, the authors provide valuable insights into the complex evolution of VOC species under the combined effects of surface emissions, chemical removal, and regional transport. The topic is certainly of interest, but the analysis does not yet sufficiently support the proposed conclusions. The existing data requires further exploration, and additional information is needed to strengthen the findings. There are also several aspects that require clarification. Below are my specific comments and suggestions.

Reply: We thank you for all the valuable comments. Each comment has been addressed in detail below and we believe the manuscript has been strengthened after the revision. We hope our replies are to your satisfaction.

**Specific Comments:**

1.My primary concern is that the current analysis does not robustly support the statement about the "complex VOC species evolution under the joint impacts of surface emission, chemical removal, and regional transport."

**1.1 Features of VOC species profiles:**

The VOC species selected for the study are appropriate, but a more thorough classification and combination would enhance the analysis. For instance, methanol and acetonitrile are largely chemically inactive and primarily derived from primary emissions. In contrast, acetaldehyde is active and predominantly the result of secondary chemical transformations.

To better address the relationship between VOC species profiles, emissions, and chemical removal, I recommend replotting Figures 4, 6, 8, and 9 to incorporate the ratios of VOC species based on their chemical reactivity (see Zhu et al., 2025) or the ratios of primary vs. secondary species (see Yang et al., 2024). Including CO and NOx profiles would also provide useful tracers for distinguishing between inactive and active species, provided the data are available.

Reply: We thank you for your suggestions. Unfortunately, there was no CO or NOx data available during the aircraft measurements. In order to incorporate the ratio analysis between primary species and secondary species as you suggested, we added an extra vertical profile of C8 aromatics-to-acetone concentration ratio in Figure 4 to address the chemical removal effect.

C8 aromatics represent primary VOC species with higher chemical reactivities that undergo fast removal in the atmosphere, while acetone represents VOC species with lower chemical reactivities and secondary formation pathways. Discussions for the vertical profile of C8 aromatics and acetone have been added in the revised manuscript (Lines 262-269):

"The ratio between VOC species is commonly applied to address the impact of chemical removal and secondary formation during transport (Yang et al., 2024b; Zhu et al., 2025). The vertical profile of the C8 aromatics-to-acetone concentration ratio was plotted in Figure 4 to demonstrate these effects. Both species can be emitted by vehicular and industrial emissions (Jiang et al., 2023; Wang et al., 2024), but C8 aromatics are more reactive, and acetone can be formed from secondary processes. The two effects both lead to a rapid decrease in the concentration ratio of C8 aromatics to acetone within the PBL before stabilizing as expected."

The purpose of Figure 6 is to compare the vertical profiles between Beijing and Baoding, and Figures 8 and 9 serve for discussions about anomalous enhancements. Thus, the proposed additional plot would not be relevant to these three figures. We would like to leave these figures unrevised. However, we have added additional discussions on the different reactivities between styrene and benzene, and the influences on the analysis in Figure 8 in Lines 360-365:

"Since styrene is more chemically reactive than benzene and thus the lifetime of styrene is much shorter than that of benzene. The ratio of styrene to benzene would decrease during transport. As shown in Figure 8d, the enhancement ratios at higher altitudes still fall within the characteristic ranges of industrial sources and are significantly larger than those of vehicular emissions. Thus, the chemical influences do not change our conclusion here".

**The revised Fig. 4:**

Figure 4. Averaged vertical profile (purple lines) of VOCs in five aerial surveys above the Beijing area in Sep. 2017 with error bar. The blue line shows the average vertical profile of C8 aromatics-to-acetone concentration ratio with error bar. The red dashed line is the average of the HPBL, with the light red area showing the variation range of one standard deviation.

**1.2 Regional Transport of VOCs:**

For the analysis of VOC regional transport, I suggest incorporating synoptic charts and diagnosing vertical velocity (including vertical transport and advection). In lines 192

and 222, the CO concentrations at ~3500 m and 2000 m could serve as effective tracers for regional transport.

Overall, the manuscript would benefit from better organization and additional data analysis to substantiate the points raised by the authors.

Reply: Thank you for your valuable suggestion. We added Figure S9 to show the wind field at 850 hPa during the periods with anomalous enhancements on Sep. 9th, 2017 and Jul. 14th, 2019. We also added additional evidence from the wind field plot to support the analysis in section 3.3 (Lines 358-360, Lines 389-392 in revised manuscript).

"The synoptic chart in **Figure S9a** shows strong northerly winds at the 850 hPa level over the aircraft survey area, suggesting that the industrial emissions were likely transported from the north."

"As shown in **Figure 9b**, winds were quite weak at 850 hPa in the aircraft survey area, suggesting that the elevated VOC concentrations were likely attributed to localized biomass burning emissions."

**Figure S9**. The synoptic charts for the flight measurements on Sep. 9th, 2017 (a) and Jul. 14th, 2019 (b). The wind field at 850 hpa was plotted. The red box indicates the area covering the trajectory of each flight as shown in Figure 1 (Route 1 and Route 6). The meteorology data were downloaded from the Global Forecast System

(https://www.ncei.noaa.gov/products/weather-climate-models/global-forecast).

2. In lines 129-132, the criteria for determining the HPBL are necessary to include, but the equation for potential temperature seems unnecessary in this context.

Reply: We thank you for your suggestion. The equation for potential temperature has been removed, and we have added additional descriptions about how HPBL was determined in the revised manuscript (Lines 148-150):

" $T_{\theta}$  first decreases with height to a minimum and then increases. The HPBL is determined as the height where  $T_{\theta}$  returns to its surface value."

3. The analysis presented in lines 230-243 is intriguing. The highest ratio of ethanol between below and above the boundary layer (BL), compared to acetaldehyde, acetone, and MEK, may reflect the more uniform vertical distribution of secondary VOCs (OVOCs) from various chemical sources. However, if the ratios of active species (e.g., styrene and monoterpenes) to inactive species (e.g., toluene and benzene) were compared, a different ratio feature of OVOCs/ethanol might emerge. These features could provide further meaningful insights that are directly related to the "complex VOC species evolution under the joint impacts of surface emission, chemical removal, and regional transport."

Reply: Thank you so much for your constructive suggestions. We have added additional discussions about the ratios of reactive species to inert species and revised the related sentences in the revised manuscript (Lines 283-290):

"C8 aromatics are quite chemically reactive, so a higher ratio suggests strong chemical removal, while for less reactive species, such as acetone, its ratio is closer to 1, indicating a weak impact of chemical reactions and potential contribution of secondary formation. Notably, the data points of styrene are clustered near the 1:1 ratio line, and occasionally, the concentrations above the PBL could be higher than those within the PBL. Since styrene is a primary species emitted at the surface, this pattern suggests a great contribution from transport, which will be discussed in **Section 3.3**."